

# The visible and hidden climatic effects on Earth's denudation

Iván Vergara[1,2*], Fernanda Santibañez[3,4], René Garreaud[1,5], Germán Aguilar[6]

[1] Center for Climate and Resilience Research (CR)2, Chile
[2] IPATEC, CONICET-UNCo, Río Negro, Argentina
[3] National University of Río Negro, IRNAD, Río Negro, Argentina
[4] National Scientific and Technical Research Council, IRNAD, Río Negro, Argentina
[5] Department of Geophysics, University of Chile, Santiago, Chile
[6] Advanced Mining Technology Center, University of Chile, Santiago, Chile

*Correspondence to*: Ivan Vergara (ivergara@comahue-conicet.gob.ar)

**Abstract.** Denudation is the opposite process of mountain uplift and plays a major role in the Earth system. Despite the research to constrain its environmental control, uncertainties remain about which are the dominant physicochemical processes at play. Here, the $^{10}$Be-derived denudation rate, encompassing time windows from $10^2$ to $10^5$ yr, was modelled in over a thousand basins across the Earth. The results suggest that water and associated life have a positive effect across their whole range, which is regulated by topography due to processes such as the energy expended by rivers on their beds, the feedback between erosion and weathering, and the transport and production rate of soils. Consequently, bioclimatic influence is weak in flat landscapes, but it could vary denudation forty times in mountain settings. It was also observed that other things being equal, water availability steepens basins, so climate also has an indirect effect acting on geological timeframes. The results can be useful for the landscape's numerical modelling and highlight the importance of climate on denudation.

## 1 Introduction

The denudation –bedrock and sediment loss from the Earth's surface– mediates the effect that climate can have on tectonics (Whipple, 2009; Hu et al., 2021; Forte et al., 2022), together with uplift drives landscape evolution (Fischer et al., 2021), and directly and indirectly influences several human activities such as agriculture, hydroelectric energy production, fishing, and water storage and consumption (Masotti et al., 2018; Li et al., 2022; Vergara et al., 2022). The denudation comprises erosion –the mechanical component– and weathering –the chemical component–. Within the first component, the following processes can be highlighted: a) fluvial erosion, that depends mostly on channel slope, which is mainly generated by spatial variability of tectonic uplift (Seybold et al., 2021); b) landsliding, that is a function of slope, lithology, soil moisture and seismicity, and usually occur upon diffusive terrains known as hillslopes (Antinao and Gosse, 2009; Campforts et al., 2022); and c) glacial erosion, which depends on basal velocity and spatiotemporal variations in meltwater drainage, and in some Earth's periods was the main denudative process (Koppes, et al., 2015; Herman et al., 2021). On the other hand, weathering generally has positive feedback with erosion and also increases with rock solubility, temperature, water availability, and vegetation, which releases acids through roots (Hinderer et al., 2013; Perron, 2017; Porder, 2019).

The difficulty of measuring denudation and the multiple physicochemical processes involved led to most of the studies that quantified its environmental control being carried





out at a regional dimension or having focused on specific processes. Regional studies
generally do not include the full range of values that controlling variables have on the Earth,
much less all the combinations of values existing between them. This can cause errors in
detecting the true effects of forcing factors, and lead to multicollinearity in statistical
approaches, which worsens the detection of the true effects (Vergara et al., 2023). On the
other hand, concentrating on specific denudation processes generated important
theoretical advances (Iverson, 2012; Roering, 2008), but it does not allow comparing the
relative weight of each one and hinders the understanding of basins' total denudation where
several processes usually coexist. These difficulties limit the knowledge about the combined
control of endogenous and exogenous forces acting in different timeframes. Here, through
statistical modelling it was identified the environmental variables that together best predict
the denudation in ~1,700 highly diverse basins around the Earth, to infer later which are
the dominant controlling processes behind (Fig. 1). Denudation was calculated from $^{10}$Be
concentration in present-day river sediment, which depends on the impact time of
secondary cosmic rays on the surface's minerals and is precisely inversely proportional to
the average denudation rate (m kyr$^{-1}$) of the upstream catchment for temporal windows
from $10^2$ to $10^5$ yr (Codilean et al., 2022).

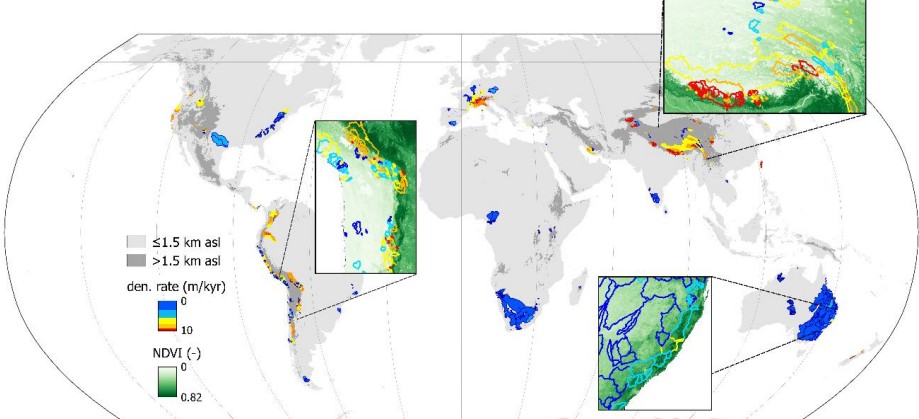

**Figure 1:** Location of the basins studied. Higher denudation can be seen on mountain ranges. Insets
with NDVI maps are regions where climate effect on denudation is evident: Hymalayas with higher
moisture southward and subtropical Andes and Great Dividing Range with higher moisture eastward
(Codilean et al., 2021).

## 2 Results

The multiple regression chosen using the automatic model selection approach has an
Adjusted R$^2$ of 78%, which would be the best physically plausible denudation prediction for
a planetary scope so far (Portenga and Bierman, 2011; Willenbring et al., 2013; Mishra et
al., 2019; Ruetenik et al., 2023) (Appendixes). The model includes the effects of terrain
slope, seismicity, lithologic hardness, cryosphere development, and the first Principal
Component between precipitation, soil moisture and vegetation development, referred as
*clim*.
$$\text{den} \propto (\text{PGA} + 1)^{0.8}\, e^{-0.004\text{lit}+0.4\text{cry}+3.2\text{sl}+0.3\text{clim}} \tag{1}$$





The predictively insignificant incorporation of the area as a covariate suggests that
[10]Be enrichment related to travel time (Carretier et al., 2009) is irrelevant for most basins.
Although seismicity and slope partly represent the same process, i.e., uplift, the model was
allowed to include both because seismicity also expresses the fracturing of rock massifs and
the triggering of landslides. The cryospheric effect includes an overestimation because
sediment coming from bedrock underlying glaciers and seasonal snowpack is shielded from
secondary cosmic rays and, therefore, does not represent a real denudation rate (Delunel et
al., 2010). In this sense, the cryospheric covariate was used to quantify the denudative
processes of this environment (Vergara et al., 2020; Zhang, et al., 2022) and to isolate the
overestimation. As an alternative method to isolate that error, a new dataset was generated
excluding basins with a plenty of solid precipitation or glacial volume, which gave analogous
results (Table S1).
The model captures the multiplicative effect between topography and water
availability that occurs on fluvial and soil erosion (Fig. 2a; see next section). However,
analysis of the raw data shows that topography and climate effects not only multiply each
other, but also increase their exponents (Fig. 2c,d). To include this phenomenon in the
model, the explicit multiplication of both covariates was added as a factor, allowing their
exponents to interact and improving the prediction by 1% (Figs. 2b & S1; Table S2).
$$\text{den} \propto (\text{PGA} + 1)^{0.8}\, e^{-0.004\text{lit}+0.4\text{cry}+3.1\text{sl}+0.1\text{clim}+0.4\text{sl clim}} \qquad (2)$$

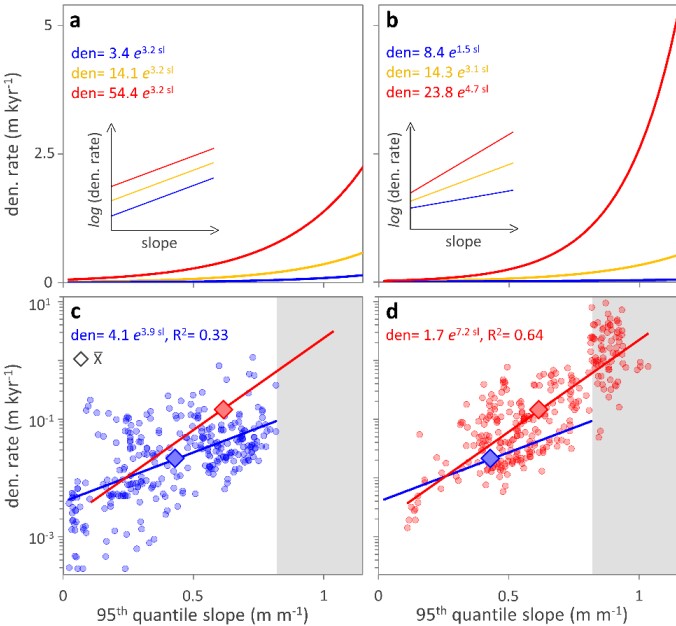

**Figure 2:** (*a*) Predicted denudation rate based on Eq. 1 as a function of slope and setting *clim* at its
minimum (blue), average (orange) and maximum (red) and the other covariates at their averages.
(b) Same as the previous panel but using Eq. 2. (*c, d*) Exponential regression between slope and
denudation rate for the 300 basins without high solid precipitation and glacial development and with
the lowest (blue) and highest (red) values of *clim*. Grey area indicates the x-axis interval without
basins with the lowest *clim* values. Note that the generation of subgroups of different sizes gives
analogous results (Table S3). Go to Fig. S2 to see the same plots, but with the roles of slope and *clim*
reversed.



## 3 Discussion

The first aspect of the model to be discussed is the joint effect –hardly separable through statistics– of precipitation, soil moisture, and vegetation. Its positive sign –even though the [10]Be method underestimates weathering (Riebe and Granger, 2013)– (Fig. 2d) indicates that denudation-promoting mechanisms, such as soil surcharge and river energy (Ferrier et al., 2013), outweigh negative ones, such as runoff obstruction by vegetation or enlarged soil cohesion caused by root anchorage (Vergani et al., 2017; Schmid et al., 2018) (Sup. Material). Furthermore, positive sign of vegetation's factor loading in $PC1_{clim}$ suggests that vegetation resulting influence is positive for the analysed temporal windows. The bioclimatic effect is widely debated precisely due to its multiple influences with opposed directions that are difficult to address together through physically based models. It is widely agreed that precipitation has a positive effect from hyper-arid to semi-arid environments, but towards wetter settings where vegetation begins to rise markedly, it was measured from a continuation of the effect's direction (Marder and Gallen, 2023), as in this study, to one or more reversals (Vergara et al., 2023; Mishra et al., 2019; Langbein and Schumm, 1958; Walling and Kleo, 1979; DiBiase and Whipple 2011; Torres Acosta et al., 2015; Chen et al., 2022). Part of the disagreement may be because many studies do not include solute load, so runoff obstruction impact on clastic load is highlighted and weathering produced by water, roots and fungi is underestimated. In addition, these studies integrate annual to decadal timeframes that do not adequately capture the recurrence of mass transport events in vegetated mountains and, consequently, the high soil production there (Mohr et al., 2023).

Another point to deepen is that the selected model represents more the denudation generated on hillslope than that generated by river network. In selecting a topographic covariate (Appendices B and C), the slope of the entire basin enters before the stream power or its morphometric solution, i.e., the Normalized Channel Steepness Index (Smith et al., 2022). This also happens for bioclimatic variables where the first one associated with runoff or streamflow enters after six related to soil moisture, vegetation or mean precipitation (Table S14). In fact, if we force a selection of models in which only topographic and climatic variables related to fluvial dynamics can enter, the chosen model explains 9% less variance (Table S4). The higher predictivity of variables related to hillslope suggests that for most basins the majority of denuded mass comes from there. The long-term denudation rate of hillslopes can be equal to that of rivers under certain evolutionary conditions of basins (Ruetenik et al., 2023; Roering, 2012; Campforts et al., 2020), but it is lower when the complete history of basins is considered, a fact that is reflected in the lower elevation of rivers relative to the surrounding mountains despite the similar uplift. Therefore, the lower denuded mass produced by rivers would be because they cover a small proportion of landscape, which is between ~0 and 3% of the non-cryospheric surface of the basins studied (Sup. Material).

The final model includes a positive interaction between slope and *clim*, whereby the effect of each is progressively amplified through the coefficient and exponent as the other increases (Fig. 2a,d). The multiplication between the covariates partly determines: a) the erosion or transport of soil, which depends on the terrain gradient and biological disturbances such as root growth (Roering et al., 2008; Chen et al., 2014), and b) the energy that rivers expend on their beds and, collaterally, the erosion they generate (Appendix B). A third process that would imply a multiplicative effect of slope and climate is the rate of soil production, which depends on the product of its moisture and the negative exponential effect of its thickness (Heimsath et al., 1997; Amundson et al., 2015). Given that, all else being equal, the frequency of soil removal and thinning increases with slope, it is possible that the rate of soil production is reflected in equations 1 and 2. Instead, the increase in



exponents generated by the interaction between topography and climate may be related to
the positive feedback between weathering -more associated with climate- and erosion -
more associated with topography-, but further research is needed on this topic (West, 2012;
Murphy et al., 2016).

The Figure 2c,d shows that the basin subgroups are separated along both axes,
suggesting not only that bioclimatic condition has an influence on denudation –Δy–, but also
that aridity may impose an upper limit on basin slopes –Δx–. Relating this possible limit with
the interaction of Figure 2b, it is estimated that, for equal and natural conditions, the
maximum variation that bioclimatic state can generate on denudation is $38^{+18}_{-12}$ times ($\equiv$
$1.1^{+0.2}_{-0.2}$ m kyr$^{-1}$) and not $109^{+108}_{-54}$, which would be calculated by assuming the existence of
extremely steep, arid basins. More importantly, the limit would indicate that a humid
climate generates steeper catchments, possibly because greater water availability produces
a denser drainage network with deeper and narrower river valleys (Rehak et al., 2010;
Harries et al., 2023). This idea is confirmed by explaining 73% of variance of basins average
slope through precipitation, lithology, cryosphere development, seismicity and elevation, so
that, in addition to the direct climate effects described, there is an indirect one associated
with denudation-promoting steepening (Fig. 3; Sup. Material; Table S6).
$$sl \propto (lit + 1)^{-0.02} (PGA + 0.1)^{0.1} pp^{0.2} (elev + 100)^{0.1} e^{0.1cry} \qquad (3)$$

The direct effects would act with a recurrence of up to a few hundreds or thousands
of years, while landscape shaping would act in time windows of hundreds of thousands to
millions of years. Finally, the similar complex relationship would occur in glacial erosion,
where the effects of terrain slope and ice thickness, i.e., climate favourability, multiply
(Cuffey and Paterson, 2010), and in turn, glacial erosion steepens terrain promoting further
erosion (Fig. 3c).

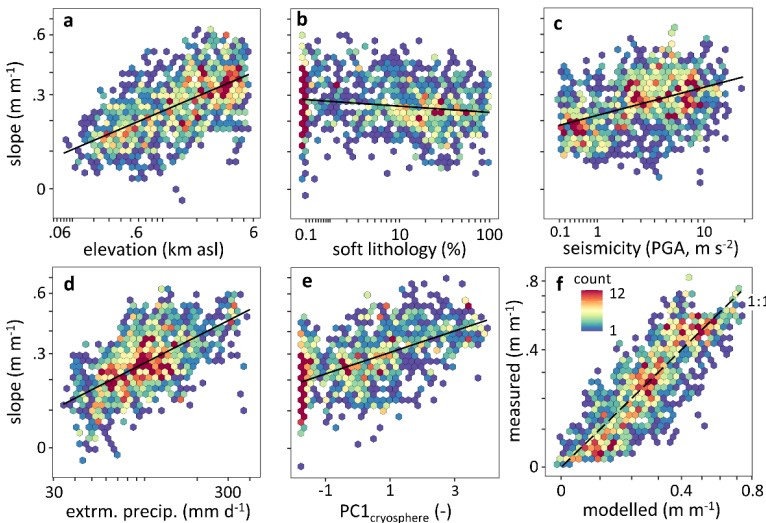

**Figure 3:** (*a*-e) Partial effects of each covariate in the regression model for the basins' average slope.
Solid lines indicate predicted fits as a function of x-axis covariate and fixing the others at their
averages. Similar relationships are obtained by predicting the 95th quantile of basin slopes (Table
S7). (*f*) Measured vs. modelled slope.





The combination of direct and indirect effects operating at different time windows
magnifies the role that climate spatiotemporal variability has on the Earth's denudation.
The suitable prediction of denudation obtained allows it to be simulated for the rest of the
Earth, to known, for example, the mass of sediments and nutrients exported to the sea. In
turn, the better understanding of the environmental processes that control denudation
could be useful to advance in the knowledge of the carbon cycle and to improve numerical
models used to study landscape evolution (Chen et al., 2014; Barnhart et al., 2020).
Appendix A: Denudation rate
A global database of [10]Be-derived denudation rates (m kyr$^{-1}$) in 4,290 basins
(Codilean et al., 2021) plus a regional database with 14 basins measured with the same
method were used (Mohr et al., 2023). In contrast to denudation rate derived from sediment
and solutes fluvial discharge, this method has minimal human disturbance and captures
large, infrequent events (Kirchner et al., 2001). To use only reliable average denudation
rates, were discarded measurements on sediment higher than 1mm and in basins smaller
than 100km$^2$ or that have lakes area plus their upstream area larger than 25% of the basin
surface. The measurements on coarse sediment were removed to avoid variations in [10]Be
concentration due to grain size and because smaller diameters would better reflect mean
denudation rate of all geomorphologic processes taking place in catchments (Carretier et
al., 2009; Aguilar et al., 2014). The small basins were filtered to avoid that [10]Be
concentration does not represent the real denudation rate due to the possible occurrence
of landslides with deep-seated failure (Yanites et al., 2009). Finally, the lake basins were
discarded because the denudation rate –calculated with spatial information of the whole
basin (Codilean et al., 2022)– does not represent the real connected area, i.e., where
sediment flux is not trapped by lakes. From the applied filters, 1,708 basins remained, but
15 more were discarded to avoid lose some candidate covariates unavailable for latitudes
greater than 60°N (see next section). The 1,693 usable basins have a median integration
time of 11.3 kyr and are around all continents except Antarctica (Table S9).
Appendix B: Controlling variables
From several global databases, 47 environmental, candidate covariates were
collected or generated. The candidate covariates were divided into the groups Topography,
Climate & Vegetation, Seismicity, Lithology and Cryosphere to reduce the computational
cost of model building and avoid collinearity (see next section). The first group includes
hydrologic variables that are computed with topographic gradient data like stream power,
while the second one includes hydrologic variables that are computed without topographic
gradient data directly like streamflow and runoff.
*Topography*
This group contain the variables: area, slope, stream power and Normalized Channel
Steepness Index ($K_{sn}$) using a concavity index equal to 0.4 and 2 thresholds of minimum
drainage area (1 and 5 km$^2$) (Hilley et al., 2019). For the last three variables, in addition to
the full averages, the 85th and 95th quantiles were calculated because steep hillslopes erode
exponentially more than flat ones (Roering, 2008), and using the average could attenuate
this signal. Also, fluvial erosion occurs above a bed shear stress threshold, so low values of
stream power and $K_{sn}$ are irrelevant (DiBiase and Whipple, 2011).
The stream power of each river reach in the basins was calculated according to the
following formula:



$$\omega = \frac{\rho\, g\, Q\, S}{b} \tag{4}$$

where $\omega$ is the stream power (W m$^{-2}$), $\rho$ is the density of water (1,000 kg m$^{-3}$), g is the acceleration due to gravity (9.8 m s$^{-2}$), Q is the maximum monthly average streamflow (m$^3$ s$^{-1}$), S is the slope (m m$^{-1}$) and b is the channel width for bankfull stage (m) that was estimated based on Q values and hydraulic geometry laws. River reaches vary in length (average 4km), but were rasterized to 250m before calculating stream power. The river network and the remaining data for each river reach were downloaded from https://www.hydrosheds.org/. In this dataset, it is assumed that a river is generated when catchment area is at least 10 km$^2$ or average streamflow is at least 0.1 m$^3$ s$^{-1}$.

While the slope was calculated with a spatial resolution of 3 arc-second (Lehner et al., 2008), stream power and $K_{sn}$ were calculated with a resolution of 15 arc-second. To ensure that the selection of slope over the other variables was due to natural causes (Discussion section), the calculated slope was replaced by another with a spatial resolution of 15 arc-second, which was also the topographic variable selected with a 1% decrease in model predictability (Table S5).

*Climate & vegetation*

As climatic variables were obtained present-day average annual precipitation (Zomer et al., 2022), 3 extreme precipitation variables (Beck, et al., 2020; Bezak et al., 2022), 2 soil moisture variables (Guevara et al., 2021), and the aridity index (AI) that is calculated by dividing precipitation with potential evapotranspiration (Zomer et al., 2022). Also was recovered average annual precipitation for the last 11.7kyr (Fordham et al., 2017), which is approximately the median integration time of the denudation rates (Table S9). Regarding hydro-climatic variables, average annual streamflow and maximum monthly average streamflow were obtained (m$^3$ s$^{-1}$) (Döll et al., 2003). In turn, these were divided by basins area to obtain runoff values (mm yr$^{-1}$).

As biologic variables it was obtained the Leaf Area index (Mao and Yan, 2019), 2 estimates of forest cover fraction (Bicheron et al., 2011; Shimada et al., 2014), the average Normalized Difference Vegetation Index (NDVI) (Leon-Tavares et al., 2021) and the C-factor, which indicates land susceptibility to be eroded by runoff (Renard et al., 1997; Borselli et al., 2008) and was calculated from the land cover map of Bicheron et al., (2011) (Table S8). Finally, the first Principal Component of the variables NDVI, AI and paleo-precipitation was calculated to have a variable that summarizes bioclimatic condition. The Principal Component captured 83% of the total variance, in fact, if in the model of Eq. 2 *clim* is replaced by any of the variables that make it up, the positive effect is maintained (Table S14).

*Seismicity*

For each basin and from Giardini et al., (2003) it was recovered the average peak ground acceleration (PGA; express in m s$^{-2}$) for a 10% probability of exceedance in 50yr, corresponding to a return period of 475yr. Also, from Pagani et al., (2020) it was obtained the average PGA express in *g* for return periods of 475 and 2,475 yr inferred from topography and not.

*Lithology*

To estimate lithologic effect, it was used the geologic map of Hartmann and Moosdorf (2012) with an average spatial scale of 1:3,750,000. Following the method proposed by





Campforts et al., (2020), to each geologic unit it was assigned an erodibility index between 2 and 12 based on its composition (Table S16). This erodibility index has a well relationship with uniaxial compressive strength at a regional dimension. In this research, values of non-igneous rocks were not weighted by their age as this data was unavailable.

The percentage of each basin with hard lithologies (metamorphic, plutonic and volcanic) and physically or chemically weak lithologies (unconsolidated sediment, evaporites, and pyroclastic and carbonate sedimentary rocks) was also calculated.

*Cryosphere*

Mean snow water equivalent (SWE; mm yr$^{-1}$), snow cover days (No. yr$^{-1}$) and frost change frequency (No. yr$^{-1}$) for each basin were obtained from Brun et al., (2022). Frost change frequency was used to represent periglacial processes such as frost-cracking. Furthermore, from Millan et al., (2022) average basal velocity (m yr$^{-1}$) of every glacier in each basin was extracted. With this dataset, the accumulated velocity of each glacier was calculated by multiplying its average velocity with its area. Then, the accumulated velocities of all glaciers in each basin were added and these values were divided by basin area. Compared to glacier area or volume, basal velocity better reflects erosion and can therefore provide more information about the $^{10}$Be-depleted sediment mass exported by glaciers (Herman et al., 2021). Finally, the first Principal Component between the four recovered variables was calculated to obtain a one that encompasses all the cryospheric processes involved in $^{10}$Be concentration.

Appendix C: Statistical approach

Once the dependent variable and the candidate covariates were collected, the environmental control on denudation rate was analysed using General Linear Model (Anderson et al., 2015). To respect linearity and homoscedasticity assumptions of this statistical method, for variables with skewness greater (less) than 0.5 (-0.5) their natural logarithms were calculated and, if necessary, translations were also be applied (by adding or subtracting a constant) so that they have a normal distribution. An automatic model selection was then performed, in which all possible models were generated by combining the candidate covariates, and the best was selected based on the Akaike Information Criterion (AIC) (Akaike, 1974), which optimises the trade-off between model prediction and complexity, i.e., variables number. To reduce computational cost of this analysis, for candidate covariates of a same group with correlations ≥0.95, those that were not the one with the highest correlation with the dependent variable were removed. Based on this rule, 30 candidate covariates were evaluated for the model building (Table S12). Here, the model selection had the restrictions of: a) for variables belonging to the same group, e.g., Cryosphere, only one could be selected to avoid collinearity and reduce computational cost, and b) the chosen regression had to be physically plausible, e.g., seismicity cannot decrease denudation. The only exception to point a) was the Topography group, where the area was not mutually exclusive with the slope variables, as they represent completely different processes. Once the best model was selected, its AIC was compared with that of another identical model, but which included a multiplication between the topographic and bioclimatic covariates, to know whether the inclusion of the interaction provided a significant improvement (Nelder, 1977).

The coefficient of determination ($R^2$) of a multiple regression increases when more covariates are introduced regardless of whether there is a real model improvement. So, predictability of the selected model was evaluated through the Adjusted $R^2$, which is independent of covariates number and therefore is a more robust metric for comparing



multiple regressions. In total, 7,543 multiple regression models were compared (Table S14). The prediction of basins slope was also performed with an automatic model selection (Sup. Material).

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

**Data availability:** All databases used are publicly accessible and their repositories are cited
throughout the manuscript and the Supplementary Material. No specific codes were
generated. Model selection and graphical representation of covariates effects were
performed using the *MuMIn* and *effects* packages of the RStudio software, respectively.

**Author Contributions:** I. V. designed the study. I. V. and F. S. conducted the analyses. All
authors interpreted the analyses and contributed to the paper writing.

**Competing interests:** The authors declare no competing interests.

**Acknowledgments:** I. V. and R. G. are partially supported by CR2 - FONDAP/ANID
1522A0001.