# Peer review of "The visible and hidden climatic effects on Earth's denudation"

_Earth System Dynamics, 2024_

## Author Comment (AC1)

**Referee #1**

This paper by Vergara et al. presents empirical models that attempt to quantify the controls on millennial-scale denudation rates using 47 candidate controlling factors.

I will focus my review primarily on the climatic controls on denudation reported in this study, because the title of the paper emphasizes these controls. Previous studies have shown that weathering rates are positively correlated with water availability throughout the full range (hyperarid to humid). Erosion by overland flow on hillslopes and by confined flow in channels has generally been shown to have a negative effect on transport rates (i.e., less vegetation cover, all else being equal, tends to result in higher transport rates, e.g., Acosta-Torres et al., 2014). Since denudation is controlled by the combination of weathering and transport, I would have expected the positive correlation of weathering with water availability and the negative correlation of transport with increasing vegetation cover to result in a complex (i.e., non-monotonic) relationship between denudation rates and water availability similar to the "humped" Langbein-Schumm (1958) relationship for short-term erosion rates and the similarly humped relationship for millennial-scale denudation rates documented by Schaller and Ehlers, 2022, https://doi.org/10.5194/esurf-10-131-2022, which Vergara et al. do not reference. I welcome any study that seeks to tease out the climatic controls on denudation. But after reading this study, I did not come away with any enhanced understanding of how climate effects denudation, nor whether this study is consistent or not with the papers referenced above.

*Reply: Thank you for your general and specific comments about the manuscript. Your review allowed substantial improvements in several sections of the text. Next, all your comments are addressed and any changes made from them are described.*

*Thank you for the suggested paper Schaller and Ehlers, (2022), we have read and cited it. It is interesting the discussion you raise about the effect of water availability on denudation. We have now extended and improved our analysis of the direct effects of water availability (lines from 117 to 137).*

*In contrast to the studies you mention (Acosta-Torres et al., 2015; Langbein-Schumm, 1958; Schaller and Ehlers, 2022), we found in our model (Fig. S1d) and in the raw data (Fig. S2c,d) that water and associated life have a positive effect over their entire range. It is important to note that Schaller and Ehlers, (2022) analyse pedons representing points in the hillslope at the basin scale, so their results are not directly comparable to those of this study, where we analyse measurements averaging areas ≥ 100 $km^2$.*

*We believe that the denudation-water availability relationship found in this research is better than those of the mentioned studies because: a) it was fitted with a greater diversity and number of basins, and b) the effects of other environmental variables that could obscure the true relationship, such as seismicity, lithology, cryosphere properties and slope, were completely isolated.*

*The fact that a large number of basins were used allowed us to avoid collinearity between covariates, and, above all, to analyse almost all the environments on Earth. In the studies with modest samples, the relationships found may be local and not representative of all the possible conditions.*

*Among the studies that you mention, Langbein-Schumm (1958) did not isolate any controlling variable before examining the relationship between precipitation and mechanical erosion, while Acosta-Torres et al., (2014) and Schaller and Ehlers (2022) partially isolated only the effect of slope before examining the relationship between precipitation and denudation. In our*

*view, it is very important to fully isolate the effects of other independent variables to ensure that they do not influence the effect of the covariate of interest. In this sense, it is important not to fit bivariate regressions in which the other independent variables are not controlled.*

*Finally, although this theory is new, there are already studies suggesting that water availability has a positive effect on denudation rate on millennial timeframes (Mohr et al., 2023; Marder and Gallen, 2023), despite the fact that vegetation obstructs runoff, which diminishes the mechanical soil erosion.*

*Marder, E., Gallen, S. F. 2023. Climate control on the relationship between erosion rate and fluvial topography. Geology, 51 (5): 424–427. https://doi.org/10.1130/G50832.1*

*Mohr, C., et al., 2023. Dense vegetation promotes denudation in Patagonian rainforests. Authorea, 1–40. 10.1002/essoar.10511846.1*

Major concerns:

1) It is clear from the presentation that the variable "clim" has a positive correlation with denudation rates. This fact may be the basis of the statement "water and associated life have a positive effect across their whole range" (line 19). What I cannot tell from the information provided is how "clim" depends on the bioclimatic variables input to it. "clim" is described as the first principal component of an analysis that includes 10+ inputs (described on lines 264-282). Lines 264-282 list 13 input datasets used to create "clim", but lines 278-282 state that 3 of these 13 datasets (i.e., NDVI, AI and paleoprecipitation) were used to derive a first principal component that was used as input. As such, the variable "clim" seems to be the first principal component of at least 13 datasets, including at least one variable that is also a principal component of multiple other datasets. Please quantify how sensitive "clim" is to each input variable. How does "clim" correlate (positively or negatively) with each input variable? Without this information, it is impossible to even begin to figure out what can be learned from this analysis or to comprehensively review this preprint. I acknowledge that some of this information may be present in one of the 16 supplementary tables. If so, that information still has to be at least summarized for the reader in the main text.

*Reply: We recognise that we have not clearly explained how the covariate "clim" was generated.*

*We collected 17 variables related to climate and vegetation, and for 3 of these 17 variables the first principal component called "clim" was calculated. Therefore, "clim" is the first principal component of 3 variables and not of 13 or 17. Among the 17 variables plus "clim" (=18), we evaluated which had the greatest capacity to predict the denudation rate, resulting in "clim" (Appendix C). Now we have described this topic better and we show in Appendix B and Table S12 how "clim" is related to the variables that make it up.*

2) Whether an empirical model is superior to another is not only a function of its goodness of fit. Models with more degrees of freedom will tend to fit any dataset better than one with fewer. Akaike's Information Criterion (Akaike, 1974) was developed to address this issue. Before the authors claim that their model is superior to others in the literature, they must report a metric that includes both the goodness of fit and the number of degrees of freedom for both their model and the alternative models. The number of degrees of freedom reflects the number of coefficients in the multivariate regression that are varied to fit the data, and also the degrees of freedom associated with the PCA used to construct the "clim" variable. I

noticed that the authors reported AIC values in their Supplementary Tables, which is laudable, but how they determined the number of degrees of freedom is unstated. There are no AIC values reported for any alternative model. As such, the authors should avoid stating that their model is superior to others in the literature based only on goodness of fit.

*Reply: We have added the degrees of freedom for each model fitted (Tables S1, S2, S5, S6, S7 and S8). We have also added the Table S3, which compares our model with the others that we mentioned. Respect the other comments we can say:*

- *The Akaike Information Criterion (AIC) should not be used to compare models that coming from different databases (i.e., models with different sample sizes or different dependent variables) (https://builtin.com/data-science/what-is-aic; Portet 2020; Symonds & Moussalli 2011). Therefore, the AIC should not be used to compare our model with the others mentioned.*
- *Instead, the adjusted $R^2$ corrects the $R^2$ based on the number of parameters and is suitable for comparing models that coming from different databases. The adjusted $R^2$ is defined as:*

$$adjusted\ R^2 = 1 - \frac{(1 - R^2)(N - 1)}{N - k - 1}$$

  *where $R^2$ is the coefficient of determination, N is the sample size and k is the number of parameters associated with the covariates (for example, a second-order polynomial has two parameters associated for a covariate).*
- *- In multiple regression, the degrees of freedom (DF) are calculated as DF= N - (k + 1). Multiple regression artificially increases its R2 as the number of parameters increases, not the number of degrees of freedom.*
- *Although the covariate "clim" was calculated from 3 variables, it is considered as a single covariate in the model.*

*Portet, S. 2020. A primer on model selection using the Akaike Information Criterion. Infectious Disease Modelling, 5, 111–128. https://doi.org/10.1016/j.idm.2019.12.010*

*Symonds, M. R. E., & Moussalli, A. 2011. A brief guide to model selection, multimodel inference and model averaging in behavioural ecology using Akaike's information criterion. In Behavioral Ecology and Sociobiology (Vol. 65, Issue 1, pp. 13–21). Springer Verlag. https://doi.org/10.1007/s00265-010-1037-6*

3) I did not see any support in the paper for the conclusion that "other things being equal, water availability steepens basins" (line 24). It's possible that this text is referring to the fact that, in equation (3), slope varies with precipitation to the 0.2 power. However, one cannot conclude from that relationship that water availability causes basins to steepen. It could just a likely be the case that steeper basins have more orographic precipitation and water availability has no causal effect on basin steepness.

*Reply: This is an interesting comment on the relationship we are proposing.*

*We acknowledge that mountain ranges increase precipitation regardless of their latitudinal range and precipitation type (frontal or convective). However, we continue to support the idea that water availability increases basin slope on scales of millions of years; therefore, equation (3) reflects a causal relationship.*

*Based on the idea that channelized water flow erodes more than diffuse water flow because it has a higher velocity due to a more defined direction and less friction with the ground, our rationale is that in basins with higher precipitation, runoff requires a smaller area to channel (A) (Montgomery & Dietrich 1989), and in turn, channels erode more because they have a higher velocity due to the greater amount of water (B) (Ferrier et al. 2013). (A) produces a denser drainage network (Rehak et al. 2010), (B) produces deeper and narrower valleys (Harries et al., 2023), and both produce a basin with a higher mean slope. These reflections were added in new lines 175-185.*

*We conducted an experiment to test whether the proposed process is more important than the positive effect generated by mountain ranges on precipitation. We fitted a regression to predict basin mean elevation instead of basin mean slope with the same data and variables as in equation (3), except for cryospheric development, as it would only reflect its strong dependence on temperature. In this case, slope became a covariate reflecting long-term uplift. In this new regression, we see that precipitation has a negative effect, which cannot be explained by orographic precipitation, but because precipitation reduces mountain height through erosion.*

$$h = pp_{extr}^{-0.9} \, (lit_a + 1)^{-0.1} \, (PGA_a + 0.1)^{0.2} \, (sl_b + 0.4)^{1.7} \, 3\,10^5 - 100$$

*This discussion was added to the Supplementary Material.*

*Ferrier, K.L., Huppert, K.L., Perron, J.T., 2013. Climatic control of bedrock river incision. Nature 496, 206–209. https://doi.org/10.1038/nature11982*

*Harries, R. M., Aron, F., Kirstein, L. A. 2023. Climate aridity delays morphological response of Andean river valleys to tectonic uplift. Geomorphology, 108804. https://doi.org/10.1016/j.geomorph.2023.108804*

*Montgomery, D. R., & Dietrich, W. E. 1989. Source areas, drainage density, and channel initiation. Water Resources Research, 25(8), 1907–1918. https://doi.org/10.1029/WR025i008p01907*

*Rehak, K., Bookhagen, B., Strecker, M. R., Echtler, H. P. 2010. The topographic imprint of a transient climate episode: The western Andean flank between 15·5° and 41·5°S. Earth Surf. Process. Landf., 35(13), 1516–1534. https://doi.org/10.1002/esp.1992*

Small issues:

1) Some of the wording needs to be improved throughout. For example: line 212: "To use only reliable average denudation rates, were discarded measurements on sediment higher than 1mm and in basins smaller than 100km2 or that have lakes area plus their upstream area larger than 25% of the basin surface." I think the word "were" should be "we". I don't know what "sediment higher than 1 mm" means. Does it mean sediment with mean diameters larger than 1 mm? There are many other examples of awkward phrasing and missing words. Another example: "This erodibility index has a well relationship with uniaxial compressive strength at a regional dimension." I think "well" is supposed to be "well-defined" in this sentence.

*Reply: Thanks to the identification of these errors, all of them have been fixed.*

2) Please provide an equation for equations (1)-(3) (not just a proportionality). Also, I don't know what some of the variables are because many are undefined. For example, in equation (1), I can guess that "den" is denudation rate and "lit" is a lithologic hardness index, etc. I searched for "PGA" and saw that it was introduced on line 285, but please make it easier for the reader by defining variables (with units) as they are introduced. I tried to figure out what some of the variables represent by going to the supplement, but that often made me more confused. For example, in trying to figure out the variables in equation (3), I went to Table S6. But some of the variable names differ between equation 3 and Table S6. For example, the lithologic hardness index seems to be referred to in Table S6 as "soft". And I couldn't figure out which variable related to precipitation in Table S6 (is it "int_daily")?

In summary, I find this analysis intriguing, but I could not, in a reasonable amount of time, determine from the information provided how the different aspects of climate (including the various aspects of precipitation (mean, seasonality, extremes) and vegetation cover) influence millennial-scale denudation rates.

_Reply_: _The completed equations and the units of measurement were added. Also, the link between the variables and their abbreviations was clarified to avoid confusion. Each variable has a unique abbreviation, which is defined in the Tables S10 and S11 depending on the database used (basins with or without significant presence of glaciers and snow)._

---

## Author Comment (AC2)

**Referee #2**

This letter presents a regression model for denudation rates based on publicly available data derived from cosmogenic nuclides. In a nutshell, my impression is that trying to compress it into a letter destroys a nice piece of work. I read the manuscript several times, permanently switching between the main text, the appendix, the supplementary text and figures and the supplementary tables. Finally, I think I understood the main ideas, but many questions remain open. To be honest, having agreed to review the manuscript was my main motivation not to give up earlier.

Written in a clear and reproducible way, this study might become a very good research paper. Since this would require a complete rewriting, I focus on a few points in the following and do not go deeper into details.

*Reply: We are pleased that you find our study a nice piece of work. Thank you for your general and specific comments. They have enabled us to improve the text considerably and are addressed below.*

*We understand the point that summarising the manuscript in a letter format can make it difficult to read, as you have to move back and forth between the main text and the supplementary figures, tables and text. However, our strategy was to produce a brief text that would appeal to a broad audience, but which would also contain all the necessary material for those who wanted to read more deeply. We know that this strategy can make reading more difficult for people who want to analyse the text in depth, but we think the balance is still positive. In this new version, we have moved some of the supplementary material into the main text in order to provide a more agile and comfortable reading experience.*

(1) The results section starts with some kind of promise that the model proposed here "would be the best physically plausible denudation prediction for a planetary scope so far." While some references are given, I did not find any serious discussion about this aspect (number of adjustable parameters, removing basins, ...). And given that the relation provided here is really the best one, what would we do with it?

*Reply: We have now added the Table S3, which compares the metrics of our model with those of the other global denudation models (number of adjustable parameters, predictive capacity, etc.).*

*Having generated a good model, we were able to give our opinion on various uncertainties in the scientific community, such as the effect of bioclimatic condition on denudation, or how the effects of bioclimatic condition and terrain slope interact. These issues were analysed in the Discussion.*

*Also, as is mentioned in the last paragraph of the manuscript, the model can be useful to estimate the denuded mass that is exported to the sea, to improve the knowledge of the carbon cycle and to understand the limitations of the current landscape evolution models, where, for example, the water effect is generally reduced to the streamflow that is estimated through the upstream area, thus assuming that precipitation does not vary spatially.*

(2) The parameters introduced in Eqs. (1) and (2) are not explained at their first occurrence and are partly not defined completely later. So it is practically impossible to recognize which of the covariates used here have the strongest effect on denudation. It is also not clear why the peak ground acceleration is taken into account in a different way than the other covariates.

*Reply: This comment was also made by the other reviewer. We now clarify what the variables introduced in Eqs. (1) and (2) are and how they have been calculated.*

*A simple way to compare the effect of each covariate is to estimate the ΔY of each panel in the Figure S1a-e. It can be seen that slope has the larger effect, seismicity, bioclimatic state and cryospheric development have similar effects, and the abundance of hard lithology has the smaller effect.*

*Peak ground acceleration (PGA) is taken into account in a different way because, unlike the other covariates, it had a higher asymmetry and so its logarithm was used (Appendix C). By solving the logarithm of the dependent variable (denudation), the logarithm of PGA is expressed as indicated in Equation (1).*

(3) The discussion section mainly addresses the limitations. After reading it, I was left with the impression that those aspect that could really deepen our understanding cannot be addressed.

(a) In the first section, it is admitted that the correlation of vegetation with other properties does not allow for a separation of its effect. In turn, it is stated in lines 122-123 that "positive sign of vegetation's factor loading in PC1clim suggests that vegetation resulting influence is positive for the analysed temporal windows." I do not understand why this is the case. To my knowledge, the loads only refer to the variability in the climatic components and not to their relation to denudation rates.

*Reply: In fact, both sentences seem to be contradictory. We remove the sentence "positive sign of vegetation's factor loading in PC1clim suggests that vegetation resulting influence is positive for the analysed temporal windows" because we are not convinced that this is true. Regarding the second comment, the loadings of a principal component analysis indicate the coefficient of correlation between the original variables and the principal components.*

(b) The second section discussed the contributions of rivers and hillslopes. In lines 145-146, it is stated that "The higher predictivity of variables related to hillslope suggests that for most basins the majority of denuded mass comes from there." I would agree that most of the denuded mass comes from the hillslopes, just owing to their larger area. Concerning the predictivity, however, it could also be that the properties used for characterizing rivers are more uncertain than those for slopes.

*Reply: We agree that data quality could have affected the predictive power of the different covariates, in fact we try to reduce this bias by performing a test where all topographic covariates had to have the same spatial resolution (Appendix B). However, the preference for hillslope-related variables over those related to the drainage network is so significant that we believe this is partly due to more sediment coming from there.*

(c) It was very difficult for me to follow the rest of this section (about the slope) and I am not sure what to learn from it.

*Reply: We now explain this section in more detail and add key points. In summary, we found that water availability increases basin slope, and given that slope increases denudation, this is an indirect effect of water availability on denudation.*

(4) Finally, I would suggest to compare the relate to those obtained by Harel et al. (2016, doi 10.1016/j.geomorph.2016.05.035). As far as I can see, these authors used basically the same data on denudation rates. As a major difference, these authors already tried to predict the parameters of the stream-power incision model for the rivers draining the respective basins. In its spirit, however, it is very similar, although the recent study is obviously more comprehensive concerning the covariates.

_Reply:_ _Thank you for the suggested paper, we have read it and compared its results with our results in the Table S3._

Overall, I think this nice work will be lost when published in its present form as a letter. So I would suggest to rewrite it and submit is as a research paper.

_Reply:_ _We have already responded to this comment._